# ALERT: Adaptive Learning with Embeddings and Reinforcement for Transparent Few-Shot Learning

## Abstract

In this paper, we introduce ALERT, a novel approach to few-shot learning that significantly enhances both the interpretability and accuracy of large vision-language models (LVLMs) in classification tasks with limited data. By utilizing the strengths of LVLMs and integrating a meta-task instruction framework, ALERT effectively transforms the traditional black-box nature of few-shot models into a transparent process. It allows for traceable and understandable reasoning. ALERT employs learnable category embeddings to emphasize unique features of each category, improving classification accuracy, and introduces a contrastive reward function within a Group Relative Policy Optimization (GRPO) training framework to enhance reasoning capabilities and training stability. Our extensive experiments across various datasets demonstrate that ALERT consistently outperforms existing few-shot learning methods, achieving state-of-the-art results. Notably, in the 16-shot setting on ImageNet, ALERT achieved an impressive accuracy of 78.74%, significantly improving on previous methods.

## 1 Introduction

In recent years, the field of computer vision has witnessed tremendous advancements, largely driven by the development of sophisticated models such as convolutional neural networks and transformers. These technologies have demonstrated extraordinary success across a spectrum of vision tasks, especially when applied to large-scale datasets. However, the challenge of limited data availability in certain scenarios has shifted focus towards few-shot learning, a burgeoning area of research. Few-shot learning involves training models to discern complex patterns and make accurate predictions even when presented with a minimal amount of labeled data.

A recent breakthrough in this domain is CLIP (Radford et al., 2021a), a paradigm-shifting approach that leverages large-scale language-image pairs for pre-training, demonstrating robust zero-shot transfer capabilities for open-vocabulary visual recognition tasks. Building on CLIP's foundation, subsequent models (Zhou et al., 2022; Gao et al., 2021; Zhang et al., 2021) have extended this zero-shot framework to few-shot classification, achieving remarkable performance across various datasets. These advancements highlight the potential of leveraging large-scale pre-training to empower networks with superior representation abilities, even when few-shot training data is scarce.

Despite these significant strides, current models often function as black boxes, producing a score for each class without offering insight into the decision-making process. This opacity is a critical limitation in applications that require transparency and understanding of model decisions. While these models achieve high accuracy, the lack of interpretability remains a persistent challenge.

In this paper, we introduce ALERT, a novel approach to few-shot learning that enhances both the reasoning capabilities and accuracy of models. Unlike traditional models, ALERT is built on the promising developments in Large Vision Language Models (LVLMs), which are designed to integrate the complementary information present in both vision and language inputs. By employing a meta-task instruction framework, ALERT transforms the traditional black-box nature of few-shot learning models into a more transparent process where reasoning can be traced and understood. As illustrated in Figure 1, our method provides more detailed analysis and identifies subtle similarities and differences, leading to more accurate classifications.

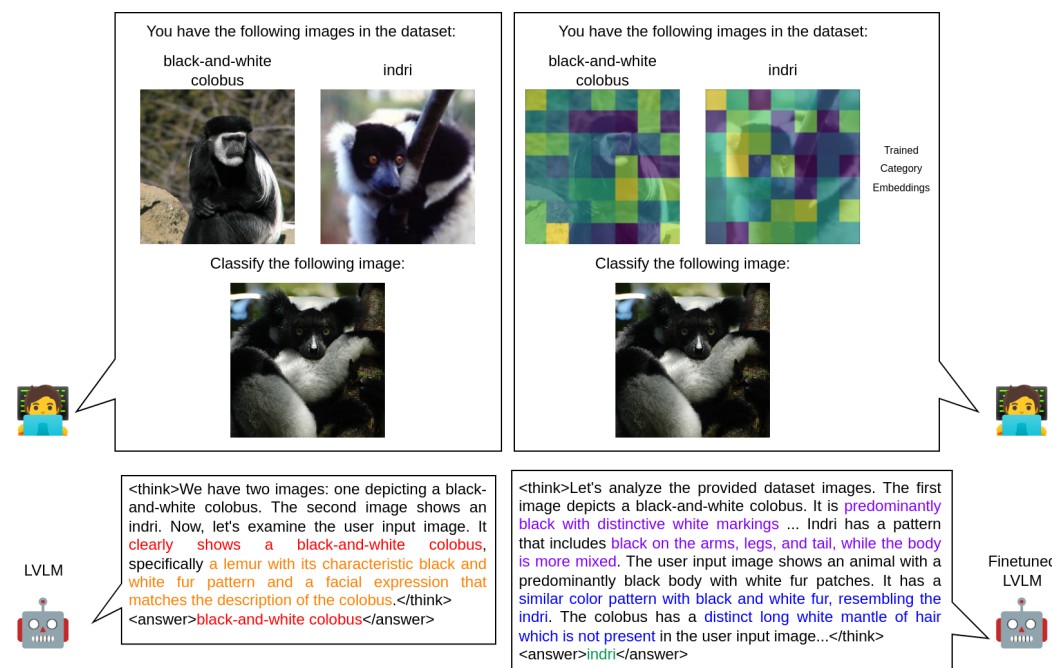

Figure 1: This example showcases our finetuned result compared with the pretrained LVLM. The pretrained LVLM (left) has premature conclusions. In contrast, the finetuned LVLM (right) provides a more detailed analysis, identifying subtle similarities and differences such as the longer white mantle of hair and mixed body pattern, leading to the correct classification.

Our contributions can be summarized as follows:

1. We propose learnable category embeddings that emphasize specific features unique to each category, improving the model's classification accuracy.

2. We introduce a contrastive reward function within a Group Relative Policy Optimization (GRPO) framework, enhancing the model's reasoning capabilities and stability.

3. We demonstrate the effectiveness of an explainable few-shot learning pipeline utilizing candidate selection and a meta-task instruction framework, which guides the model in generating clear and logical reasoning for classification tasks.

## 2 RELATED WORKS

### 2.1 FEW-SHOT LEARNING

Few-shot learning is a classification paradigm that focuses on the ability of models to generalize from a limited number of training examples, often as few as one or two per class. Unlike traditional machine learning approaches that require large amounts of labeled data, few-shot learning aims to mimic the human ability to learn new concepts quickly with minimal data. Typically, this is achieved by leveraging prior knowledge gained from related tasks or datasets, enabling the model to adapt rapidly to new tasks with minimal additional training.

Prominent strategies include meta-learning, which trains models to adapt quickly to new tasks (Chen et al., 2021; Finn et al., 2017; Jamal & Qi, 2019; Li et al., 2021), and metric learning, which constructs a feature space where classification is based on proximity (Snell et al., 2017; Sung et al., 2018; Vinyals et al., 2016). However, these approaches often lack interpretability, making it difficult to understand their decision-making process.

More recently, vision-language models like CLIP (Radford et al., 2021a), pre-trained on web-scale data, have shown significant promise. Adaptation techniques such as prompt tuning in CoOp (Zhou et al., 2022) and lightweight adapters in CLIP-Adapter (Gao et al., 2021) further enhance their performance on downstream few-shot tasks. Despite these advancements, a common criticism remains: the lack of transparency in how these models arrive at their decisions, which is crucial for applications requiring explainability.

Our approach addresses these weaknesses by integrating a meta-task instruction framework within a Large Visual Language Model (LVLM) to provide interpretability alongside high accuracy.

## 2.2 LVLM INSTRUCTION TUNING

LVLMs represent a significant advancement in vision-language integration. Inspired by successful instruction tuning in LLMs, recent work has focused on enhancing LVLMs with instruction-following data to improve performance across diverse tasks. For instance, LLaVA-1.5 (Liu et al., 2023) improved on Visual Question Answering, InstructBLIP (Dai et al., 2023) enhanced zero-shot capabilities, and Gemma 3 (Team et al., 2025) refined dialogue interactions through such tuning.

Our approach stands out by incorporating learnable category embeddings during the LVLM tuning process, unlike typical methods that rely only on text and images. This innovation enhances classification accuracy and interpretability, allowing the model to make more nuanced comparisons in few-shot learning scenarios. By combining instruction tuning with category embeddings, our method fully utilizes the LVLM's potential for precise and interpretable few-shot classification.

## 2.3 GROUP RELATIVE POLICY OPTIMIZATION (GRPO)

GRPO (Shao et al., 2024) is a reinforcement learning technique that has been instrumental in advancing the reasoning capabilities of LLMs. This method was notably utilized in the DeepSeek-R1 model (DeepSeek-AI, 2025), which showcased the potential of pure RL strategies in fostering complex reasoning patterns within LLMs. A pivotal discovery from DeepSeek-R1 is the ability of GRPO to induce the emergence of chain-of-thought reasoning and "aha moments," where models actively re-evaluate and correct their reasoning processes.

The GRPO approach involves generating a group of outputs for each training sample from an existing policy. It calculates advantages using pre-defined reward functions for each output, guiding the policy update towards more refined reasoning. This optimization strategy enables models to develop reasoning capabilities by reinforcing outputs that demonstrate superior logical coherence and accuracy.

In our work, we adapt the GRPO optimization strategy from DeepSeek-R1 to enhance few-shot learning tasks. This adaptation involves proposing a contrastive reasoning reward function to improve training stability and accuracy. By incorporating this reward function, we aim to bolster the model's ability to discern subtle differences and similarities within few-shot learning scenarios, thus enhancing its interpretability and performance.

## 3 METHODOLOGY

### 3.1 CATEGORY CANDIDATE SELECTION

In the initial stage of our few-shot learning pipeline, we can employ any established few-shot learning method for selecting potential category candidates. For each given image, we utilize a few-shot learning model as a lightweight draft model to generate a logit for every category within the dataset. Each logit quantifies the likelihood that the image belongs to a particular category. We then select the top-$k$ categories with the highest logits as candidate categories. These selected categories are incorporated into the instruction prompt for the Large Visual Language Model (LVLM) during the second stage of our pipeline. In this stage, the LVLM is tasked with choosing the most likely category for the image during the inference process. This methodology effectively reduces task complexity and enhances the LVLM's self-consistency (Liu et al., 2025) by narrowing down the range of possible categories, thereby facilitating more accurate classification.

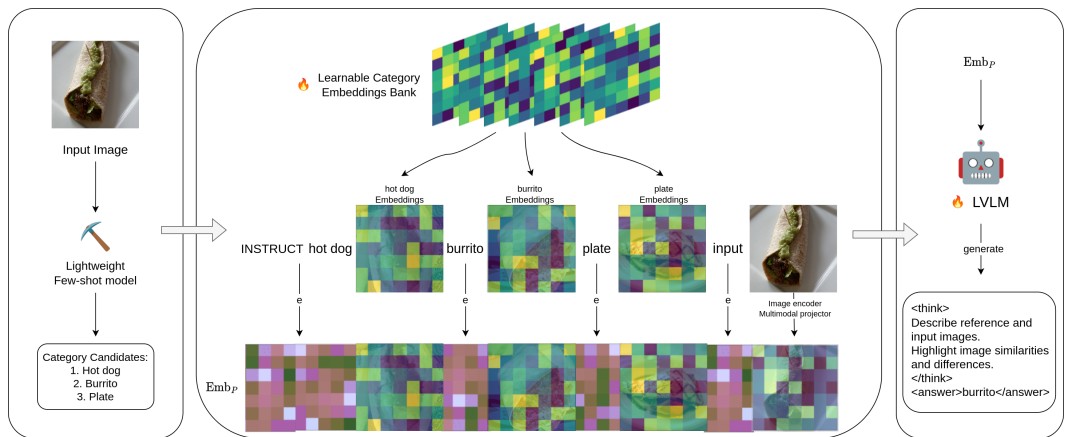

Figure 2: This diagram illustrates our ALERT few-shot learning pipeline. It begins with an input image, which is processed by a lightweight few-shot model to select category candidates. The embeddings of these categories are then combined with the input image embedding and text embeddings to form a meta-task instruction prompt. This prompt is fed into the LVLM, which generates reasoning and a classification answer.

## 3.2 TRAINABLE CATEGORY EMBEDDING

Each category within the dataset has its own trainable 2D embedding matrix. Mathematically, the embedding for category $c_i$ can be denoted as $\text{Emb}_{c_i} \in \mathbb{R}^{m \times d}$ where $m$ is the number of tokens per image embedding and $d$ is the embedding dimension. For each category, we first extract image embeddings from the few-shot training images using the LVLM's image encoder. These image embeddings are then averaged to form an initial representative embedding for the category. This averaged embedding is subsequently passed through the LVLM's multimodal projector to create the initialized category embedding. These initialized category embeddings are stored in a category embeddings bank, and we let the model access and update them during training.

## 3.3 META-TASK INSTRUCTION

We have developed a meta-task instruction framework to prompt the LVLM into generating responses. This framework requires reference categories from the dataset to be included within the instruction prompt. When classifying a given image, the LVLM selects the most likely reference category from a set of reference categories, denoted as $\{r_{c_1}, r_{c_2}, \ldots, r_{c_k}\}$, where $k$ represents the number of reference classes. The meta-task instruction prompt $P$ is structured as follows:

$$T_{\text{instruct}} \oplus r_{c_1} \oplus \text{Emb}_{c_1} \oplus \cdots \oplus r_{c_k} \oplus \text{Emb}_{c_k} \oplus \text{Emb}_{\text{input}}$$

where $\oplus$ signifies the concatenation operation. $T_{\text{instruct}}$ is the text instruction designed to guide the LVLM in classifying the given image. The terms $\text{Emb}_{c_i}$ represent the category embeddings of the reference classes, which are retrieved from the category embeddings bank. $\text{Emb}_{\text{input}} \in \mathbb{R}^{m \times d}$ represents the image embedding of the input image, which is extracted using the LVLM's image encoder and multimodal projector. The complete meta-task instruction prompt is detailed in Appendix A.

During training, we randomize the number of reference categories provided in the instruction prompt. One of the dataset category labels is the ground-truth image label $c_{gt}$, while other labels are sampled from all available dataset categories based on the similarity between the ground-truth category images and those of other categories.

We begin by calculating the cosine similarity between the L2-normalized average image embeddings from the few-shot training images of each category, resulting in a logit that quantifies the similarity

between the ground-truth category $c_{gt}$ and each other category in the dataset:

$$l_{c_{gt},c_b,k} = \left( \frac{\sum_{i=1}^{k} f_{c_{gt}}^i / k}{\left\| \sum_{i=1}^{k} f_{c_{gt}}^i / k \right\|} \right)^T \left( \frac{\sum_{i=1}^{k} f_{c_b}^i / k}{\left\| \sum_{i=1}^{k} f_{c_b}^i / k \right\|} \right)$$

Here, $l_{c_{gt},c_b,k}$ represents the similarity logit for the $k$-shot learning scenario between category $c_{gt}$ and another category $c_b$. The term $f_{c_{gt}}^i$ denotes the image embedding extracted from the $i$-th image in category $c_{gt}$, and $f_{c_b}^i$ denotes the image embedding from the $i$-th image in category $c_b$.

Reference categories are sampled based on the computed weights:

$$w_{c_{gt},c_b,k} = \frac{\exp(l_{c_{gt},c_b,k}/\tau)}{\sum_{j=1}^{C} \exp(l_{c_{gt},c_j,k}/\tau)}$$

In this equation, $\tau$ is a temperature parameter (less than 1) used to adjust the likelihood of selecting categories that are more similar to the ground-truth label. A lower $\tau$ value increases the difficulty of the training samples by making the selection of similar categories more pronounced.

During inference, the reference categories are the top-$k$ classes from the category candidate selection.

### 3.4 TEXT GENERATION FROM EMBEDDING

To process the instruction text part of our meta-task instruction prompt $P$, we begin by tokenizing the instruction text. These text tokens are then converted into their corresponding embeddings, denoted as $\text{Emb}_P$. Throughout this process, we retain all category embeddings and the input image embedding in their original form. The embedding sequence is structured as follows:

$$\begin{aligned}
\text{Emb}_P = [&e(T_{\text{instruct},1}), \ldots, e(T_{\text{instruct},I}), \\
&e(r_{c_1,1}), \ldots, e(r_{c_1,C_1}), \text{Emb}_{c_1}, \ldots, \\
&e(r_{c_k,1}), \ldots, e(r_{c_k,C_k}), \text{Emb}_{c_k}, \text{Emb}_{\text{input}}]
\end{aligned}$$

where $\{T_{\text{instruct},1}, \ldots, T_{\text{instruct},I}\}$ represent tokens derived from $T_{\text{instruct}}$, $\{r_{c_1,1}, \ldots, r_{c_1,C_1}\}$ are tokens from $r_{c_1}$, $\{r_{c_k,1}, \ldots, r_{c_k,C_k}\}$ are tokens from $r_{c_k}$, and $e(\cdot)$ is the token embedding function for mapping tokens to their respective embeddings.

The next token logits can be obtained by

$$H = \text{Transformer}(\text{Emb}_P)$$
$$z = W \cdot h / T$$

where $H \in \mathbb{R}^{M \times d}$ is the matrix of the last hidden states for all tokens up to position $M$, $h = H[M,:]$ is the last hidden state of position $M$, $W$ is the parameter of the language model head and $T$ is the sampling temperature.

Finally, the next token is sampled from the probability distribution derived from the temperature-scaled logits $z$. This sampled token is then converted into its corresponding embedding and appended to the sequence, allowing the LVLM to continue generating the response based on the initial instruction prompt and input. This process repeats for each subsequent token until an end condition, such as a specific token indicating the end of the response, is met.

### 3.5 GRPO REWARD FUNCTIONS

Reward functions used in GRPO reinforcement learning score the text completions generated by a model based on pre-defined criteria. We employ four reward functions in GRPO training:

1. Strict Format Reward Function: This function verifies whether the completion adheres to the required format by enclosing reasoning within `<think>` and `</think>`, and the predicted label within `<answer>` and `</answer>`. This approach is inspired by the format reward function used in DeepSeek-R1 (DeepSeek-AI, 2025).

2. Soft Format Reward Function: This function evaluates whether the completion at least partially aligns with the reasoning format.

3. Accuracy Reward Function: This function checks if the predicted label enclosed within `<answer>` and `</answer>` precisely matches the ground-truth label.

4. Contrastive Reasoning Reward Function: We use an LLM as a judge to give a score to the generated reasoning text. The score is based on whether the generated reasoning is able to identify similarities and differences between the provided image and dataset images, highlight relevant details, and ensure the class decision is justified by the identified similarities and differences. The full prompt used for this LLM judge is provided in Appendix B. This reward function can improve the RL training stability.

## 4 EXPERIMENTS

### 4.1 EXPERIMENTAL SETTINGS

#### 4.1.1 DATASETS

Following previous works (Zhou et al., 2022; Zhang et al., 2021; 2023), we perform few-shot experiments on 11 publicly available datasets: ImageNet (Deng et al., 2009), StandfordCars (Krause et al., 2013), UCF101 (Soomro et al., 2012), Caltech101 (Fei-Fei et al., 2004), Flowers102 (Nilsback & Zisserman, 2008), SUN397 (Xiao et al., 2010), DTD (Cimpoi et al., 2014), EuroSAT (Helber et al., 2019), FGVCAircraft (Maji et al., 2013), OxfordPets (Parkhi et al., 2012), and Food101 (Bossard et al., 2014). Specifically, we train ALERT under the few-shot setups with 1, 2, 4, 8, 16 shots and test on full test splits in all datasets.

#### 4.1.2 TRAINING DETAILS

We use the instruction-tuned Gemma 3 (Team et al., 2025) 4B model as our pretrained LVLM. To enhance training efficiency and reduce costs, we load the model in 8-bit precision and freeze both the LLM and visual encoder, employing LoRA (Hu et al., 2021) to fine-tune the model's adapter for 4 epochs.

The learning process utilizes a cosine learning rate scheduler with a base learning rate of $5 \times 10^{-6}$ and a warm-up ratio of 0.01. Optimization is performed using the AdamW optimizer (Loshchilov & Hutter, 2019), with a weight decay of 0.1, $\beta_1$ set to 0.9, and $\beta_2$ set to 0.99, ensuring stable convergence.

During training, we randomize the number of reference categories, ranging from 2 to 8. For reference category selection, we set the temperature parameter $\tau = 0.1$ to increase the difficulty of the training samples.

We use vLLM library (Kwon et al., 2023) to speed up the text generation process in GRPO training. We configure the number of generations per training sample to be 6 and set the maximum completion length for each generation to 512 tokens. The sampling temperature $T$ is set to 0.9. We evaluate text completions using the following reward function scoring scheme to guide the model's learning process:

1. Strict Format Reward Function: Assigns a score of 0.5 if the completion strictly adheres to the specified format, with reasoning enclosed in `<think>` tags and the predicted label in `<answer>` tags; otherwise, the score is 0.

2. Soft Format Reward Function: Provides a score of 0.5 for partial compliance with the required format, scoring 0 if these criteria are not met.

3. Accuracy Reward Function: Awards a score of 2 if the predicted label matches the ground-truth label exactly, otherwise 0.

4. Contrastive Reasoning Reward Function: Uses the Gemma 3 4B LLM to evaluate reasoning quality, scoring from 1 to 10, normalized to a 0 to 1 scale.

Figure 3: **Performance (%) Comparison on ImageNet.**

| Shot | 0 | 1 | 2 | 4 | 8 | 16 |
|------|-----|-----|-----|-----|-----|-----|
| Zero-shot CLIP | 60.33 | - | - | - | - | - |
| Gemma 3 4B | - | 64.71 | - | - | - | - |
| Linear-probe CLIP | - | 22.17 | 31.90 | 41.20 | 49.52 | 56.13 |
| CoOp | - | 57.15 | 57.81 | 59.99 | 61.56 | 62.95 |
| CLIP-Adapter | - | 61.20 | 61.52 | 61.84 | 62.68 | 63.59 |
| VT-CLIP | - | 60.53 | 61.29 | 62.02 | 62.81 | 63.92 |
| Tip-Adapter-F | - | 61.32 | 61.69 | 62.52 | 64.00 | 65.51 |
| CALIP-FS | - | 61.35 | 62.03 | 63.13 | 64.11 | 65.81 |
| CaFo | 62.99 | 63.80 | 64.34 | 65.64 | 66.86 | 68.79 |
| **ALERT** | - | **76.95** | **77.80** | **78.63** | **78.69** | **78.74** |

Table 1: **Quantative Performance (%) Comparison on ImageNet.**

### 4.1.3 BASELINES

In line with previous studies (Zhang et al., 2023), our baseline comparisons include zero-shot CLIP (Radford et al., 2021a) and several recent few-shot learning methods, such as Linear-probe CLIP (Radford et al., 2021b), CoOp (Zhou et al., 2022), CLIP-Adapter (Gao et al., 2021), VT-CLIP (Qiu et al., 2023), Tip-Adapter-F (Zhang et al., 2021), CALIP-FS (Guo et al., 2022), and CaFo (Zhang et al., 2023). These models are evaluated based on their accuracies on each dataset.

In addition to these established baselines, we introduce the pretrained Gemma 3 4B model as a new baseline. For the Gemma 3 4B model, instead of using trained category embeddings, we randomly select one image per category from the training dataset and utilize its image features extracted by Gemma's image encoder. This setup is considered a 1-shot setting. The model then generates reasoning and an answer for each test image, and we use the answer as the prediction to compare against the ground-truth class label.

### 4.2 PERFORMANCE

### 4.2.1 ON IMAGENET

We compare ALERT performance with our baselines on the most representative ImageNet in Figure 3 and Table 1. Throughout our experiments, unless otherwise specified, we chose to use the top-2 candidates selected from CaFo during the category candidate selection stage. In the 16-shot config-uration, ALERT attains an accuracy of 78.74%, notably surpassing the zero-shot CLIP baseline at 60.33% and the one-shot Gemma 3 baseline at 64.71%. Compared to other few-shot learning mod-els, ALERT consistently demonstrates superior accuracy, with enhancements ranging from 14.46% to 20.92% over CaFo across various shot settings. Impressively, ALERT with just 1 shot outperforms all other methods in every setting.

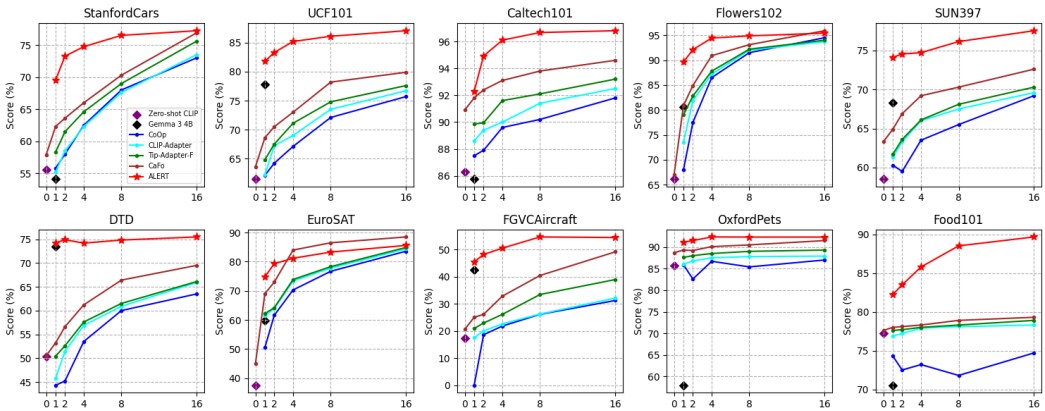

Figure 4: Performance (%) Comparison on 10 Datasets.

| Datasets | Source | Target | |
| --- | --- | --- | --- |
| | ImageNet | -V2 | -Sketch |
| Zero-shot CLIP | 60.33 | 53.27 | 35.44 |
| Gemma 3 4B | 64.71 | 64.87 | 57.69 |
| CoOp | 62.95 | 54.58 | 31.04 |
| CLIP-Adapter | 63.59 | 55.69 | 35.68 |
| CALIP-FS | 65.81 | 55.98 | 35.37 |
| Tip-Adapter-F | 65.51 | 57.11 | 36.00 |
| CaFo | 68.79 | 57.99 | 39.43 |
| **ALERT** | **78.74** | **76.70** | **69.12** |

Table 2: **Distribution Shift (%) Comparison.** We train the models on Source dataset and test on Target datasets.

### 4.2.2 ON OTHER DATASETS

To further evaluate ALERT's robustness across different scenarios, we tested it on an additional 10 datasets, as detailed in Figure 4. These datasets, which cover a wide range of semantic categories such as real-world scenes, detailed textures, and satellite-captured landscapes, allowed us to assess how well ALERT adapts to diverse data characteristics. The results demonstrate that ALERT consistently outperforms other few-shot learning models.

For instance, on the OxfordPets dataset, ALERT achieved an accuracy of 92.3%, surpassing the performance of CaFo, which reached 91.5%. In the Caltech101 dataset, which includes diverse object categories, ALERT excelled with an impressive accuracy of 96.8%, outperforming CaFo's 94.6%. The model's superiority also extends to more specialized domains. On the DTD texture dataset, ALERT achieved an accuracy of 75.5% (vs. CaFo's 69.5%), and on Food101, it scored 89.7%, significantly outperforming CaFo's best result of 79.3%.

While ALERT demonstrates competitive performance across most datasets, it underperforms compared to CaFo on EuroSAT. Specifically, ALERT's accuracy tops at 85.6% in the 16-shot scenario, whereas CaFo achieves 88.5%. This discrepancy indicates that while ALERT excels in general few-shot learning tasks, there is room for improvement in handling satellite image classifications.

### 4.2.3 DISTRIBUTION SHIFT

To further assess ALERT's robustness to distribution shifts, we conducted experiments where the model was trained on a Source dataset and then evaluated on Target datasets. In our study, as pre-

sented in Table 2, we used ImageNet as the Source dataset, while ImageNet-V2 and ImageNetSketch served as the Target datasets.

The results demonstrate ALERT's superior performance over all baselines under distribution shifts. It achieved 76.70% accuracy on ImageNet-V2 and 69.12% on ImageNet-Sketch, showcasing its strong adaptability. These improvements indicate that ALERT effectively inherits valuable prior knowledge from the pre-trained LVLM, enabling it to generalize well to new and unseen data distributions and maintain high performance.

## 4.3 ABLATION STUDIES

### 4.3.1 CATEGORY EMBEDDINGS BANK

To evaluate the impact of trainable versus frozen category embeddings, we conducted an ablation study focusing on the Category Embeddings Bank. In this study, we compared the performance of ALERT using trainable category embeddings against a version with frozen embeddings. The results, as shown in Table 3, indicate that the trainable embeddings significantly enhance the model's performance, achieving an accuracy of 78.74% compared to 69.81% with the frozen embeddings. This improvement underscores the ability of trained embeddings to effectively capture category-specific features and distinguish them from features of other categories, thereby leading to more accurate classification outcomes.

| Embedding Type | Accuracy (%) |
| --- | --- |
| Frozen Category Embeddings | 69.81 |
| Trainable Category Embeddings | 78.74 |

Table 3: Performance (%) Comparison of ALERT with Frozen vs. Trainable Category Embeddings.

### 4.3.2 CONTRASTIVE REASONING REWARD FUNCTION

We conducted an ablation study to assess the necessity of the contrastive reasoning reward function in our model's training process. Without this reward function, we observed significant training instabilities after approximately 0.5 epochs, leading to accuracy ranging only between 65.8% and 76.4% on ImageNet. By including the contrastive reasoning reward function, we enhanced training stability and achieved a much higher accuracy of 78.74%. Throughout our experiments, we experienced no stability issues when the contrastive reasoning reward function was used, even with extended training periods. This highlights the critical role of the contrastive reasoning reward function in stabilizing the training process and improving the model's overall performance.

| Configuration | Accuracy (%) |
| --- | --- |
| Without Contrastive Reward | 65.8 - 76.4 |
| With Contrastive Reward | 78.74 |

Table 4: Performance (%) Comparison of ALERT with and without the Contrastive Reasoning Reward Function on ImageNet.

## 5 CONCLUSION

We introduced ALERT, a novel few-shot learning approach that enhances both the interpretability and accuracy of LVLMs. By combining a meta-task instruction framework with learnable category embeddings and a contrastive reasoning reward function within a GRPO framework, ALERT makes the classification process transparent and traceable. Extensive experiments across 11 datasets show that ALERT consistently outperforms existing methods, achieving state-of-the-art accuracy. Ablation studies confirmed that our proposed trainable embeddings and contrastive reward function are critical for enhancing performance and training stability.

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

## A  Meta-task Instruction Prompt

In our meta-task instruction prompt shown in Listing 1, the placeholders `<Reference category label>`, `<Reference category embedding>`, and `<Image embedding>` are replaced with specific values during the process. Specifically, `<Reference category 1 label>` is replaced with the category label $r_{c_1}$, `<Reference category 1 embedding>` is

**Listing 1** Prompt for our meta-task instruction

```
You are an image classifier. You are given a dataset containing images
    with their corresponding labels. When a user provides an input image,
     your task is to classify it into one of the existing classes. Follow
     these steps:
1. Carefully examine and describe the dataset images, noting key details
    and characteristics of each class.
2. Analyze and describe the user input image in detail.
3. Compare the user input image with the dataset images:
    - Identify and highlight similarities between the input image and the
        images of each class.
    - Identify and highlight differences between the input image and the
        images not belonging to the chosen class.
4. Use your observations to determine the most appropriate class for the
    user input image, ensuring your reasoning is clear and logical. Then
    provide your classification decision.

The reasoning process and answer are enclosed within <think></think> and
    <answer></answer> tags, respectively, i.e., <think>reasoning process
    here</think><answer>answer here</answer>.
Dataset images:
Class name: <Reference category 1 label> <Reference category 1 embedding>
Class name: <Reference category 2 label> <Reference category 2 embedding>
...
User input image: <Image embedding>
```

**Listing 2** Prompt for scoring the reasoning

```
Evaluate the student's reasoning for classifying an image based on the
    following criteria and provide a score from 1 to 10:
- Identification of similarities between the given image and the dataset
    images of the chosen class.
- Identification of differences between the given image and the dataset
    images not belonging to the chosen class.
- Identification of relevant details in both the given image and the
    dataset images.
- The student claims the class only after clearly highlighting the
    similarities and differences.

The student's reasoning:
<reasoning>

Output only the score (1-10) without formattings and explanations.
```

replaced with the corresponding category embedding $Emb_{c_1}$, and `<Image embedding>` is replaced with the image embedding $Emb_{input}$. These substitutions ensure that the LVLM receives all necessary information to accurately classify the input image by comparing it with the reference categories and their embeddings, which are included in the instruction prompt.

## B CONTRASTIVE REASONING JUDGE PROMPT

The contrastive reasoning judge prompt shown in Listing 2 is designed to evaluate the quality of the LVLM's reasoning when classifying an image. It instructs the evaluator to score the reasoning on a scale from 1 to 10 based on several criteria: the identification of similarities between the given image and dataset images of the chosen class, the identification of differences with images not belonging to the chosen class, and the identification of relevant details in both the given image and the dataset images. The prompt emphasizes that the LVLM should only claim the class after clearly highlighting these similarities and differences. In this prompt, the placeholder `<reasoning>` is replaced with the reasoning text generated during GRPO training, allowing for an objective assessment of the classification process.

## C  VISUALIZATION OF CLASSIFICATION AND REASONING

Figure 5 provides additional visualizations that illustrate the detailed reasoning process of our model during the classification task on examples from the ImageNet dataset. This figure showcases two examples, demonstrating the model's capability to handle varying numbers of reference categories effectively.

## D  ADDITIONAL ABLATION STUDIES

### D.1  CATEGORY SELECTION COUNT

To understand the impact of the number of category candidates on ALERT's performance, we conducted an ablation study varying the number of top candidates selected during the classification process. The selection of top category candidates is crucial for narrowing down the possible classes, which can significantly affect the model's ability to make accurate predictions. We evaluated ALERT using different numbers of top category candidates: top-2, top-3, top-4, and top-8.

| Number of Top Candidates | Accuracy (%) |
| --- | --- |
| Top-2 Candidates | 78.74 |
| Top-3 Candidates | 76.63 |
| Top-4 Candidates | 75.23 |
| Top-8 Candidates | 72.85 |

Table 5: Performance (%) of ALERT with Different Numbers of Top Category Candidates on ImageNet.

The results are summarized in Table 5. The study indicates that selecting the top-2 candidates results in the highest accuracy of 78.74%. This suggests that focusing on the most relevant categories during inference allows the model to maintain high precision without the distraction of too many options.

### D.2  CATEGORY EMBEDDINGS INITIALIZATION

| Initialization Method | Accuracy (%) |
| --- | --- |
| Random | 76.20 |
| Averaging Method | 77.63 |
| Proposed Method | 78.74 |

Table 6: Performance (%) Comparison of Different Initialization Methods for Category Embeddings.

In our experiments, we explored different initialization strategies for category embeddings to assess their impact on the performance of our model. We tested two additional methods alongside our primary approach: random initialization and an averaging method that combines embeddings after the image encoder layer and the multimodal projector layer. The random initialization involves setting the category embeddings to random values, while the averaging method computes the mean of embeddings obtained from both the image encoder and the multimodal projector layers.

Our results in Table 6 show that the choice of initialization method can affect the model's performance, albeit slightly. The random initialization achieved an accuracy of 76.20%, while the averaging method resulted in a 77.63% accuracy. These findings indicate that while initialization strategies can influence the outcomes, their overall effect is relatively minor compared to other factors in the model design and training process.

Reference categories:

Arctic fox      Alaskan tundra wolf      Samoyed      polar bear

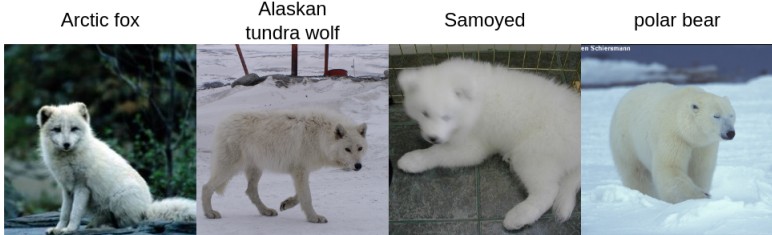

Input image:

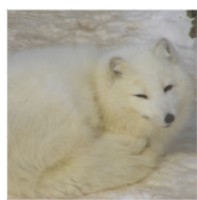

<think>Let's analyze the dataset images first. We have an Arctic fox, an Alaskan tundra wolf, a Samoyed, and a polar bear. The Arctic fox is characterized by its thick, dense white fur and relatively small size – it looks like a fluffy, compact animal. The Alaskan tundra wolf is larger with a more rugged appearance and darker coloring around the eyes and muzzle. The Samoyed is distinctly characterized by its extremely fluffy white coat, giving it a cloud-like appearance. Finally, the polar bear is the largest of the group, with a thick layer of white fur and a distinctive orange nose. Now, let's examine the user input image. It shows a white animal with thick fur, lying down. This animal has rounded ears, a relatively small nose, and a gentle expression. Comparing the user input image to the dataset images, it most closely resembles the Arctic fox. Both have thick white fur and a compact, rounded appearance. The user input image also portrays a lying-down posture, similar to the Arctic fox in the dataset.</think>
<answer>Arctic fox</answer>

Reference categories:

agama      desert grassland whiptail lizard      green iguana      alligator lizard

frilled-necked lizard      Gila monster      chameleon      Komodo dragon

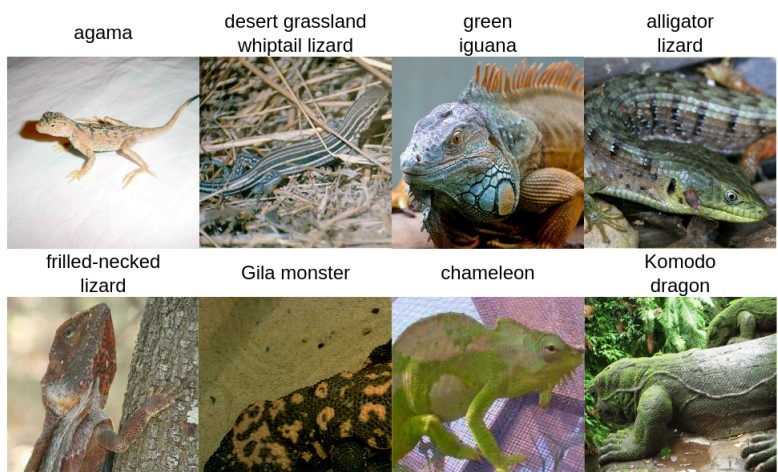

Input image:

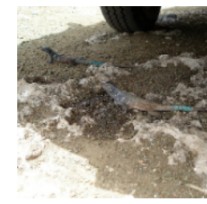

<think>Let's analyze the dataset images first. We have several lizard species represented: agama, desert grassland whiptail lizard, green iguana, alligator lizard, frilled-necked lizard, Gila monster, chameleon, Komodo dragon. The images show various colors, patterns, and sizes of lizards. The agama image shows a brightly colored lizard on a rock, while the whiptail lizard is a smaller, slender lizard on bare ground. The green iguana is large and heavily armored, and the alligator lizard is similar in size and shape to the whiptail, but with different coloration. The frilled-necked lizard has a distinctive frill around its neck, the Gila monster is pink and patterned, the chameleon is a small, camouflaged lizard, and the Komodo dragon is a very large, powerful reptile. Now, let's examine the user input image. The image shows two lizards on a rocky surface, similar to the whiptail lizard. They are of similar size and coloration. Considering these similarities, the most likely classification for the user input image is 'desert grassland whiptail lizard'.</think>
<answer>desert grassland whiptail lizard</answer>

Figure 5: Additional visualizations demonstrating candidate selection with 4 categories (top) and 8 categories (bottom). The above examples are from the ImageNet dataset. The images of the reference categories are for illustration purpose only. We use trained category embeddings instead of individual images.

