# OpenReview forum: "ALERT: Adaptive Learning with Embeddings and Reinforcement for Transparent Few-Shot Learning"
_ICLR.cc/2026/Conference — Submitted to ICLR 2026_

### Official Review · Reviewer_BQB9 · 2025-10-16

**Soundness:** 3
**Presentation:** 2
**Contribution:** 2
**Rating:** 2
**Confidence:** 5

**Summary:**

This paper proposes ALERT, a novel framework for transparent few-shot learning that enhances both the accuracy and interpretability of Large Vision-Language Models. The method combines three key innovations: Learnable Category Embeddings: category-specific embeddings that capture discriminative visual features; Meta-Task Instruction Framework: reformulates classification as an instruction-following task where the model generates explicit reasoning enclosed in special tags; Group Relative Policy Optimization with a Contrastive Reasoning Reward – reinforcement learning that encourages logically consistent and explainable reasoning. Experiments on 11 public few-shot datasets show that ALERT consistently surpasses existing approaches such as CoOp, Tip-Adapter-F, and CaFo.

**Strengths:**

Integrating reinforcement learning and LVLM instruction tuning for few-shot learning is a fresh and creative combination. The authors also conduct extensive evaluation across 11 datasets and various shot settings.

**Weaknesses:**

1. For previous methods, e.g., Zero-Shot CLIP, CoOp, they compute the cosine similarity between image and text embeddings, which may not be viewed as "black-box" methods. As for the proposed method, it uses a modern LVLM compared to previous methods. The authors' claim that by utilizing GRPO to encourage LVLM to output reasoning chains for better performances, as the solution to the "black-box" problem, is not fair.
2. From Table 1 and Figure 4, it seems the proposed method encounters a bottleneck as the number of shots grows. However, other methods, e.g., Tip-Adapter-F, CaFo, show steady improvements.
3. The most suitable baseline method for the proposed method, as far as I consider, should be zero-shot Gemma-4B. However, the authors do not provide results on this setting.
4. The visualizations, e.g., reward curves, about the overall RL training process are missing.

**Questions:**

See weakness.

---

### Official Review · Reviewer_BZU9 · 2025-10-17

**Soundness:** 3
**Presentation:** 3
**Contribution:** 3
**Rating:** 6
**Confidence:** 3

**Summary:**

The paper proposes ALERT, a few-shot learning framework combining learnable category embeddings, meta-task instruction tuning, and GRPO reinforcement learning. It aims to improve accuracy and interpretability in vision-language models. The paper’s contribution lies more in smart engineering synthesis and empirical validation than in foundational novelty, which isn't a weakness. Overall, it is a solid and well-written paper that provides a practical advance in interpretable few-shot learning

**Strengths:**

S1. Good integration of LVLMs and reinforcement learning. Integrating instruction-tuned LVLMs with reinforcement learning-based reasoning is a fresh angle for few-shot learning.

S2. Strong empirical results across multiple datasets.

S3. Clear method and ablation studies.

**Weaknesses:**

W1. Relies on proprietary Gemma 3 model; reproducibility is limited, and limited clarity on computational cost, training time, or efficiency trade-offs versus simpler baselines.

W2. Interpretability claims are mostly qualitative. The “transparency” claim is mostly qualitative (no quantitative measure of interpretability).

W3. If the method scales to other domains, it could have a high impact.

**Questions:**

Q1. Can you provide quantitative interpretability metrics (e.g., fidelity, agreement with ground-truth rationale)?

Q2. Can you discuss training efficiency and scaling cost explicitly?

Q3. Would you consider using a smaller open model (e.g., LLaVA-based) for reproducibility?

---

### Official Review · Reviewer_dGPm · 2025-10-31

**Soundness:** 3
**Presentation:** 3
**Contribution:** 3
**Rating:** 4
**Confidence:** 3

**Summary:**

This paper introduces ALERT (Adaptive Learning with Embeddings and Reinforcement), a novel approach for few-shot learning. It aims to address the "black box" nature of current models by significantly enhancing both the accuracy and interpretability of Large Vision-Language Models (LVLMs). The ALERT method makes the model's classification process transparent and its reasoning traceable.

**Strengths:**

1. The primary strength of ALERT is that it significantly enhances both the classification accuracy and the interpretability of Large Vision-Language Models (LVLMs) in few-shot tasks.
2. The model transforms the traditional "black-box" nature of few-shot learning into a transparent process. It allows for reasoning that is traceable and understandable, providing more detailed analysis and identifying subtle similarities and differences to make correct classifications.
3. ALERT consistently outperforms existing few-shot learning methods across various datasets, achieving state-of-the-art results.

**Weaknesses:**

1. High Computational Cost and Complexity. The ALERT pipeline is very complex and computationally intensive: Stage 1: Requires an additional lightweight FSL model (like CaFo 1) for category candidate selection. Stage 2: Relies on a large LVLM (Gemma 3 4B). GRPO training itself is very time-consuming, requiring the generation of multiple (6 in this paper) outputs for each training sample. Most critically, the paper's proposed "contrastive reasoning reward function" needs to call another LLM to act as a "judge" to assign a score. This means that in every step of RL training, an additional LLM inference is required, which will lead to astonishingly high training costs. The authors should more fully discuss this trade-off between accuracy/interpretability and computational cost in the paper.
2. The success of the entire pipeline depends on the performance of the lightweight model in Stage 1. If the true category is not included in the top-k candidates (in the experiments, $k=2$ performed best), the LVLM will be unable to make the correct classification. This paper does not fully discuss this potential "error accumulation" effect. If the initial model's recall is insufficient, ALERT's performance ceiling will be severely limited.
3. About the performance on EuroSAT: The authors candidly note that ALERT underperforms compared to CaFo on the EuroSAT dataset. While this is a minor issue, it is worth further discussion. Why does this complex model, which excels on 10 other datasets, fail at satellite image classification? Does this imply that the world knowledge and reasoning capabilities brought by the LVLM are not applicable or are even misleading in this domain (remote sensing)?

**Questions:**

See weaknesses.

---

### Official Review · Reviewer_4S3J · 2025-10-31

**Soundness:** 2
**Presentation:** 3
**Contribution:** 2
**Rating:** 4
**Confidence:** 4

**Summary:**

This paper presents ALERT, a new approach to few-shot learning that enhances both accuracy and interpretability in large vision–language models (LVLMs). ALERT introduces a meta-task instruction framework that makes the model’s reasoning more transparent, addressing the typical “black-box” limitations of LVLMs. It further boosts performance through learnable category embeddings and a contrastive reward objective trained with Group Relative Policy Optimization. Experiments across multiple datasets show that ALERT sets a new state of the art in few-shot classification, including 78.74% accuracy on ImageNet with only 16 shots, surpassing existing methods by a substantial margin.

**Strengths:**

The novelty of ALERT lies in combining interpretability and performance in few-shot learning for large vision–language models, leading to more explainable and accurate classification with limited data.

ALERT demonstrates

•	Improved interpretability – The meta-task instruction framework makes the model’s reasoning process transparent.

•	High performance – ALERT achieves state-of-the-art accuracy in few-shot classification, notably outperforming prior methods on benchmarks like ImageNet.

•	Innovative training design – The use of learnable category embeddings and a contrastive reward within the GRPO framework enhances both reasoning stability and feature discrimination.

**Weaknesses:**

There are various shortcomings. If addressed the paper would be improved.

•	Complex training framework may increase implementation difficulty and computational cost. While the framework is described a figure outlining the pipeline would help the reader understand how the different components interact. Figure 2 does not do that.

•	Limited evaluation scope — there are many parameters and while the values used are specified, it is unclear how they were set. Additionally the evaluative scoring function hides where the different methods excel. It would be interesting to break this down to understand where the gains are made. For some of the datasets where the score is increasing at 16 shots, it would be interesting to see how it continues to increase for larger shot values. Is there a value for which the other methods surpass ALERT?

•	The work makes a valuable contribution and builds effectively on current advances. However, including a discussion of remaining challenges and possible avenues for future research would strengthen the paper and highlight its long-term potential.

**Questions:**

In the methodology, you say that you "can employ any established few-shot learn- ing method for selecting potential category candidates." Is there evidence of this? Can you justify this claim?

Can you please explain how each of the parameters were set in the experimental section and the effect of different parameter values?

Can you show the breakdown of the scores from the scoring function for each of the methods to demonstrate exactly where the gains are made?

Can you do some experiments that show up to 50 shots on datasets where the score is still increasing? It appears that ALERT plateaus early while others are still increasing.

The paper claims transparency and traceability. Can you please describe the meaning of traceability and discuss how ALERT enhances traceability?

---

### Official Review · Reviewer_qvVa · 2025-10-31

**Soundness:** 2
**Presentation:** 1
**Contribution:** 1
**Rating:** 2
**Confidence:** 5

**Summary:**

ALERT is a two-stage few-shot pipeline built on an LVLM. First, a candidate selector (e.g., CaFo) provides top-k candidates as the reference. Second, the LVLM receives an instruction-style prompt that concatenates text tokens, reference category embeddings, and the input image embeddings, then generates a chain-of-thought and a final label. Training uses GRPO-style reinforcement learning.

**Strengths:**

1. This paper follows a good structure.
2. Promising results are obtained in the studied benchmarks.

**Weaknesses:**

1. The motivation is not clearly articulated.

(1)The claim that *ALERT transforms few-shot learning from a black-box into a transparent process* is not convincing. The method still relies on an auto-regressive LVLM whose internal computations remain opaque; producing a chain-of-thought-style text does not, by itself, establish transparent and faithful process, as explanations can be plausible yet misleading[1]

(2)Moreover, existing LVLMs already support few-shot learning with explicit reasoning traces via instruction tuning or in-context learning (e.g., Flamingo[2], GPT-4V[3], Qwen-VL[4], DeepSeek-VL[5]), so the claim that “ALERT addresses the weakness… to provide interpretability” in Line 114 is not well supported.


2. Novelty and discussion in related work are limited

(1) For learnable category embeddings, category prototypes or caches are well studied in CLIP-based few-shot adaptation (e.g., CLIP-Adapter, TIP-Adapter). I suggest authors discuss more about what is architecturally or algorithmically new beyond moving the idea into an LVLM context.

(2) For the meta-task instruction framework, the term meta might be misleading, since ALERT formats inputs as instructions and decodes with a standard LVLM, which is closer to instruction tuning or in-context learning than to meta-learning.

(3)For contrastive reward, using an LLM-as-judge to grade reasoning is widely used, and GRPO was introduced to forego the critic model in PPO[6]. Please acknowledge related work.


3. The method description needs clarity.

(1) Section 3.4 can be greatly simplified. This section essentially restates standard LVLM decoding: tokenize text; interleave text tokens with visual tokens; run a Transformer; project the last hidden state to vocabulary logits; sample next token. I suggest that authors reduce to a concise description and cite prior LVLMs (e.g., Flamingo) for the interleaved-token formulation

(2) I also suggest the author provide a complete training algorithm pipeline, especially including input, output, and reward/loss to make the contribution easier to follow.

4. Some implementation details are missing.

(1) How are learned category embeddings parameterized and updated?

(2) What are m and d for your backbone?

(3) LoRA hyperparameters;

(4) Inference decoding details

5. The experiment demonstration is weak.

(1) Baselines may be outdated and comparisons unfair. Table 1 mixes substantially different base systems. Results of CaFo are from 2023; meanwhile, strong recent LVLMs that natively support few-shot reasoning (e.g., Qwen-VL, DeepSeek-VL, GPT-4V) are absent. I suggest that authors add such baselines and compare with more recent SoTAs.

(2) CaFo (2023) uses base models such as GPT-3, CLIP, DINO, and DALL-E. If one of the key motivations is to pursue higher accuracy, I suggest authors demonstrate how the original CaFo framework performs with current stronger base models and show whether ALERT still adds value beyond this simple modification.

(3) Why is Gemma-3 (2–16-shot) “–” in Table 1?

(4) ALERT relies on reference categories (top-k from CaFo). It might be unfair to compare with baselines that do not use candidate information or that use randomly selected candidates.

(5) I am also concerned about how ALERT performs when the ground-truth is not in top-k provided by CaFo

(6) The author should report and compare the number of trainable parameters for different baselines and provide training/inference throughput, GPUs, wall-clock time compared to baselines, especially because GRPO + LLM-as-judge can be costly and an additional LVLM is introduced.

(7) In Fig. 3, ALERT appears to saturate after 4-shot while baselines continue to improve. The authors did not analyze this.

(8) In Fig. 4, the ALERT fails to surpass CaFo at larger shots on EuroSAT. Can the authors provide more analysis?

(9) More ablation studies are needed: per-class trainable embeddings v.s. shared projector on category embedding; reward weights and specific removals; number of generations per training sample; and max-length ablations.


6. Please ensure your tables follow ICLR’s style: “The table number and title always appear **before** the table.”

[1] Language Models Don’t Always Say What They Think: Unfaithful Explanations in Chain-of-Thought Prompting. In NIPS, 2023

[2] Alayrac, Jean-Baptiste, et al. "Flamingo: a visual language model for few-shot learning." *Advances in neural information processing systems* 35 (2022): 23716-23736.

[3] Yang, Zhengyuan, et al. "The dawn of lmms: Preliminary explorations with gpt-4v (ision)." *arXiv preprint arXiv:2309.17421* (2023).

[4] Wang, Peng, et al. "Qwen2-vl: Enhancing vision-language model's perception of the world at any resolution." *arXiv preprint arXiv:2409.12191* (2024).

[5]Lu, Haoyu, et al. "Deepseek-vl: towards real-world vision-language understanding." *arXiv preprint arXiv:2403.05525* (2024).

[6] Shao, Zhihong, et al. "Deepseekmath: Pushing the limits of mathematical reasoning in open language models." *arXiv preprint arXiv:2402.03300* (2024).

**Questions:**

See Weaknesses

---

### Comment · Area_Chair_JBfw · 2025-11-20
**To AI Review**

Dear authors,

I would like to share an important reminder regarding the review process. Recently, we have noticed that some reviewers may be using AI tools to help generate their reviews. This can lead to low-quality or inaccurate feedback, which is unfair to authors who deserve careful and thoughtful evaluations.

To help maintain fairness, I kindly ask for your assistance: If you believe a review you received was partly or fully generated by AI, and you have some evidence (for example: unusual writing style, clear factual mistakes, AI-detector results, repeated generic sentences, etc.), please feel free to contact me directly.

I will review any evidence you provide and, if appropriate, adjust the weight of the reviewer’s evaluation so that it does not negatively affect your submission. Thank you for helping us keep the review process fair and responsible. Your understanding and cooperation are greatly appreciated.

Best regards,

AC

---

### Comment · Area_Chair_JBfw · 2025-11-27

Dear Reviewers and Authors,

As we are approaching the rebuttal deadline, I would like to share a gentle reminder with everyone.

For authors:
If you have not yet submitted your rebuttal, please make sure to do so as soon as possible. Submitting very close to the deadline may reduce the chance for reviewers to read and respond in time, which could affect the discussion phase.

For reviewers:
If a rebuttal has already been submitted for your assigned paper, I encourage you to take a moment to read it and, where appropriate, provide a brief response or update your evaluation. Of course, this is not meant to pressure anyone into changing scores, it is simply to ensure that all reviews remain well-informed before final decisions.

Thank you all for your time and effort in keeping the review process smooth and constructive.

Warm regards,
AC

---

### Meta-Review · Area_Chair_65x2 · 2025-12-07

**Summary:**

The proposed method lacks sufficient novelty, and the model itself comes with high computational cost and complexity. Some of the baseline comparisons appear outdated, and the experimental validation is not comprehensive enough to convincingly support the claims. Notably, the paper’s interpretability “transparency” motivation is mostly qualitative and not convincingly justified, as chain-of-thought outputs can be unfaithful and do not constitute a measurable transparent process. Moreover, the multi-stage pipeline (candidate selection + GRPO with multiple generations + LLM-as-judge) likely incurs substantial training/inference overhead, yet the paper does not provide sufficient efficiency reporting or scalability analysis to justify this trade-off.

The submission is borderline in its current form and still has substantial room for improvement in terms of novelty, clarity, and empirical validation. Therefore, my final decision is reject.

**Reviewer Concerns:**

The authors have not submitted the rebuttal.

**Reviewer Scores:**

Scores: 2, 4, 4, 6, 2

The authors have not submitted the rebuttal.

---

### Decision · Program_Chairs · 2026-01-26

Reject